# Online Unproctored Clinical Examinations: The Perceptions and Performance of Final Year Dental Students

**DOI:** 10.3390/dj10110200

**Published:** 2022-10-25

**Authors:** Laura Gartshore, Mark Jellicoe, Joanne Bowles, Girvan Burnside, Luke J. Dawson, Vince Bissell

**Affiliations:** School of Dentistry, University of Liverpool, Liverpool L3 5PS, UK

**Keywords:** COVID-19, open book examinations, take home examinations, dental education, assessment, perceptions

## Abstract

Background: Take home, or open-book, examinations (OBE) are designed to be completed at a location of student choice, whilst providing comprehensive assessment of learning outcomes. Supporters of OBE refer to their authenticity, in that they reflect real-world practice where use of external resources is routine and encouraged. A contrasting view is that efficient practice requires a solid base of knowledge upon which to draw. The aim of this evaluation was to elicit learners’ perceptions of the open-book, unproctored examination approach; we sought student views on authenticity, assessment preparation, use of resources, and anxiety. Methods: Quantitative and qualitative data were gathered using an online, self-administered survey. We sought to determine the correlation between student views and examination performance via consideration of final examination marks. Results: Heightened anxiety levels tended to increase assessment preparations and were found to be inversely related to learners’ perceptions that the OBE was an authentic test. An inverse relationship was seen between learners’ OBE examination performance and consulting resources during the examination. Examination marks were not significantly related to endorsement of continued online delivery of learning, time spent preparing for OBE in comparison to other types of assessment, greater anxiety than usual, perceptions of test authenticity, or experiencing a supportive test environment. Conclusions: The results of this study may inform curriculum and assessment development, learning and teaching practices, and support student voice and experience.

## 1. Introduction

A recent systematic review by Bengtsson suggests various drivers for the adoption of “Take Home Examinations (THEs)”, which include the desire to assess higher-order cognitive skills, the massification of higher education with accompanying changes in student learning habits, and the need for a more comprehensive assessment of learning outcomes [1]. Bengtsson’s review considers THEs to be an extension of proctored Open Book Examinations (OBE); THEs are “*an exam that the students can do at any location of their choice, it is non-proctored and the time limit is extended to days (rather than hours…)*” [1]. A hallmark of the open-book assessment type, is that they may permit, and in some cases are designed to encourage, learners to consult resources of their choice. More recently, the necessity of COVID-19 could be added to this list of drivers [2], as the pandemic led to the rapid development of a range of alternative assessment methods [3], including OBEs. Educators aim to adopt research-informed approaches in developing curricula [4], however, the literature that compares OBE approaches to conventional assessment methods in medical and dental education is somewhat limited [5], The requirement to assure a range of stakeholders of assessment validity contributes to caution when considering changes in assessment type, which often favours maintaining the status quo. That said, tentative assurance is offered in research that may support change, as comparable outcomes have been reported when comparing OBEs and traditional methods [5]. Nevertheless, it is noted that such understandings were not established against the backdrop of a global pandemic [1,5].

In response to COVID-19, and in common with educators from allied health professions, in 2020 the School of Dentistry at The University of Liverpool, was forced to redesign and deliver an online finals assessment approach at pace. Globally there has been recognition that this driver favoured expedience over good design [3]. A marked and largely untested change in assessment method, at short notice, was an unanticipated shift. In particular, change often holds with it the very real possibility of raising levels of student anxiety and destabilising learners’ feelings of competence [6], which could, in turn, have adverse impacts on academic outcomes [7,8]. This may be exacerbated when the window to introduce and prepare learners for such a change is limited.

The design of existing assessment blueprints at the school aims to ensure that competency is holistically evaluated through triangulation across a range of assessments including a significant amount of longitudinal work-based data available for each student, supplemented by data from simulated-environment assessments such as an OSCE. The assessment of applied knowledge that underpins the competencies relies on data from traditionally proctored single best answer (SBA) examinations. In the circumstances of the pandemic, a pragmatic solution was required that managed the loss of non-work-based, face to face assessments. Therefore, a replacement was designed that could:be delivered online and unproctored (the timeline was considered to be too short to obtain and quality assure a method of remote proctoring), and which would avoid compromising the existing, limited item bank.test the same attributes, such as the ability to synthesise clinical information and exercise clinical judgement for diagnostic, treatment planning and patient management purposes.

An OBE with multiple short answer format questions was designed, based on clinical vignettes with appropriate clinical images. The OBE permitted learners to consult resources, but the question design anticipated that students would not do so extensively in the given time constraints.

Drawing on the literature examined, we aimed to elicit and explore relationships between final examination marks and student views on:Self-reported (a) anxieties associated with OBE, (b) time preparing for the OBE, and (c) time spent consulting resources during the OBEStudent perceptions (a) about the authenticity of the assessment method as a test of their competence and (b) of support derived from the learning environmentWhether learners endorsed the continuation of online (a) assessment and (b) associated delivery.

Our decision to make this change was driven by necessity, however, the situation also presented an opportunity to rethink how assessment, in this case, OBE, contributes to our assessment blueprint. Importantly, this also allowed us to explore the impact of the introduction of this novel method on learners. In this latter regard, we note a recent call by Zagury-Orly and colleagues to do just this [2] and the need to remain agile and change assessment strategies to maintain an optimal approach [4]. The current paper presents data from a *post hoc* evaluation of student perceptions of the open book approach.

## 2. Materials and Methods

### 2.1. Participants

The eligible population consisted of final year students of the undergraduate and graduate entry dentistry (BDS) programme (*n* = 74). Female students account for 65% of the eligible undergraduate population. All participants in the target population responded.

### 2.2. Materials and Methods

#### 2.2.1. The Examination

Open-book, online, short answer question (SAQ) assessment formats were designed. The University directed a doubling of time available for completion of each assessment. This measure allowed students to manage the open-book nature of the assessments, whilst also addressing practical concerns about internet connectivity and the necessary extra time for those students with additional needs during assessment. The assessments were administered unproctored using the University’s virtual learning environment (Canvas). The novel assessment arrangements were communicated to students in writing, and during synchronous online meetings held with senior staff. Further preparations included a practice assessment before the final examination to enable students to system test the software, and understand the requirements of the open-book approach and format. The Undergraduate Programme Director and the TEL (technology-enhanced learning) team were available throughout the assessment period to address issues arising, including, for example, from students who might have suffered significant disruption to their internet connectivity. Arrangements were made to enable more than one sitting of examinations so that those who were unwell or self-isolating on the date of the assessments were able to sit on an alternative date. Students were invited to submit queries regarding the assessments anonymously following the practice and system tests. A response to these queries was generated in FAQ format for dissemination to all students via Teams in advance of the actual examination.

#### 2.2.2. The Evaluation

A novel online survey tool was designed to address the objectives of the study. Items were generated following a literature review and discussion with senior staff responsible for student experience and evaluation, which included a psychologist. After item pooling, item reduction was completed to minimise survey length and therefore participant burden. The tool was pre-tested via senior staff responsible for student learning and assessment and subsequently piloted by the student co-chair for the Staff-Student Liaison Committee. Following consideration of pilot feedback, the tool was edited to reduce duplication. Several items exploring the research question were defined. These concepts included items that considered learner anxieties, for example ‘I felt more anxious about exams than I usually do’ and assessment preparation, for example, ‘How much time did you spend preparing for online exams compared with your usual amount of preparation?’. Several items spoke to learners’ perceptions that the OBE was a fair test of their competence, for example, ‘The exams provided me with an opportunity to demonstrate my clinical competence’. The survey also asked learners to consider whether the online learning environment was supportive, for example, ‘How was our communication with you during this period?’, and whether assessment arrangements were well managed, for example, ‘The online assessment arrangements were appropriate at a time when face to face assessments could not take place’. Twelve survey items employed a five-point Likert scale response format to indicate the strength of agreement, probability and or quantity i.e., much more time vs. much less time. In each case higher scores, i.e., 5, are related to a greater level of agreement or higher probability. Four additional survey items employed free-text responses, these items included ‘Do you have any suggestions to help us improve online assessment?’ and ‘What have we done that best supported your online learning experience?’.

Ethical approval was gained from the University of Liverpool (May 2020). This included permission to identify and track participants responses to enable comparison of individual perceptions with examination performance (marks obtained). Responders were identifiable only to a research statistician who anonymised the responses for analysis by the staff researchers. Demographic data were not gathered as it was not relevant to the research question and may have led to responder bias and suboptimal response rate if students had concerns that they might be identified by the staff researchers. If a student identified themselves in the free-test responses, the principal investigator anonymised these data before analysis occurred.

Data were gathered via an online survey using Microsoft Forms, with a link being posted to students on Microsoft Teams and via email by the TEL team. This administration approach aimed to minimise any perception of social desirability reporting in participants responses. Participation was voluntary and learners completed the survey online at a time of their choosing, free in the knowledge that a decision not to participate would have no bearing on assessment outcomes. The survey was administered following completion of the assessment period but before the Board of Examiners decision on final examination outcomes were made and released to learners. This timing meant that responses were close to the time of the assessment and participants responses were not influenced by knowledge of their assessment results. The response rate was 100%.

Participant responses were available for analysis immediately following the closure of the survey. Data were exported into a csv file. Where a participant failed to respond to an item, these responses were not included in the analysis. Quantitative analyses were conducted in JASP [9]. First, in the quantitative phase of analysis, we examined descriptive data and identified that data were non-parametric. Next, a dimension reduction technique, Principal Component Analysis, tentatively examined an assumption that responses to seven survey items systematically addressed two underlying concepts. This was confirmed, which supported grouping these items and deriving mean scores for two sub-scales. The first including four items addressed ‘the OBE as an authentic test of competence’ (4 items; α = 0.80). The second included three items and addressed learner perceptions that a ‘supportive learning environment’ existed (3 items; α = 0.69). These subscales indicated good and reasonable levels of internal consistency respectively. Further information on the Principal Component Analysis can be obtained by contacting the corresponding author. However, given the distribution, Spearman’s correlation analysis examined the relationships between participant survey responses and final examination marks, in pursuit of the research aims. Participants free-text responses to the open-ended item ‘Do you have any suggestions to help us improve online assessment?’ were analysed using NVivo using an approach approximating thematic analysis [10]. Nearly four-fifths of participants responded to this question.

## 3. Results

### 3.1. Quantitative Data Analysis

After running the preparatory descriptive analyses, data were employed in pursuit of the research aims. Both descriptive data and correlation effect estimates are reported (Table 1). A narrative description of results follows. Largely, these summarise the significant relationships depicted in Table 1, except where we determine that non-significant relationships support understanding; however, all relationships can be seen in Table 1 for completeness.

Exploratory correlation analysis indicated that when learners reported heightened anxieties about OBEs they tended to increase their assessment preparations (*r_s_* = 0.55, *p* < 0.001). These heightened anxiety levels were also found to be inversely related to learners’ perceptions that the OBE was an authentic test allowing them to demonstrate competence (*r_s_* = −0.43, *p* < 0.001), and that learners endorsed the continuation of online assessment (*r_s_* = −0.27, *p* < 0.05), meaning that learners who reported lower levels of anxiety were more likely to endorse the authenticity of the assessment and were inclined to endorse the continuance of OBEs. Supporting this, when learners considered OBEs an authentic test of competence, they were more likely to welcome OBEs in future (*r_s_* = 0.42, *p* < 0.001). Potentially enhancing these perceptions, learners who reported that the learning environment was supportive, also endorsed the authenticity of OBE (*r_s_* = 0.23, *p* = 0.052) although this result marginally exceeded the 0.05 alpha threshold typically used to indicate significance. These results also indicated that learners who supported online delivery of learning also endorsed the continuation of online assessment (*r_s_* = 0.39, *p* < 0.001).

The marks achieved in the examination ranged from 59% to 94% (Figure 1). The Hofstee passmark was 59%. This cohort’s performance in the examination did not appear particularly unusual in the experience of the authors, although no detailed comparisons with other cohorts or assessments has been undertaken.

When considering the final examination marks, an inverse relationship was seen between learners OBE assessment performance and consulting resources during the exam (*r_s_* = −0.25, *p* = 0.035) and in relation to continuing with online OBEs (*r_s_* = −0.27, *p* = 0.019). This suggests that students achieving marks towards the higher end of the range tended to report that they consulted resources less often during the assessment than their colleagues who did less well in comparison. In addition, higher-performing students were also more comfortable with continuing online OBE as an assessment method. It is noteworthy that examination marks were not significantly related to endorsement of continued online delivery of learning, time spent preparing for this assessment in comparison to other types of assessment, greater anxiety than usual, perceptions of test authenticity or experiencing a supportive test environment when compared with usual. Visual inspection indicated a largely non-parametric distribution of the participant responses to survey items, confirmed in each case by a significant Shapiro-Wilks test *p* < 0.05 (Figure 2).

### 3.2. Qualitative Data Analysis

Three themes emerged from participants’ responses to the open-ended question ‘Do you have any suggestions to help us improve online assessment? These were inter-related and included ‘assessment completion time’, ‘practice opportunities’ and ‘acceptance’. Learners diverged in their views on assessment completion time with some calling for more time during the assessment to allay practical concerns:


*“Submit via plagiarism software instead or with online invigilation like the medical school and give a longer time to do it. Lots of anxiety worrying about internet connection dropping!”*


This response included tacit acknowledgement by a student that the security of the test was crucial to provide assurance of test validity and acceptance. However, another student addressing assessment completion time called for shorter assessments, but at the same time increasing their quantity, due to fatigue:


*“I found the exam very long (took me almost 4 hours) and although I thought the exam was relevant and tested my knowledge thoroughly, my eyes were starting to strain by the end of it and I was feeling very tired. Perhaps having shorter exams but more of them may help resolve this problem.”*


Addressing the theme acceptance, this split may indicate that an optimal assessment length was achieved, which balanced the needs of individual learners. However, practically, acceptance for some was paired with a suggestion for greater variety in question format:


*“I feel providing mixed formats of questions in future exams rather than 100% one type will allow students a nice middle ground.”*


This may also indicate that aspects of the question design, paired with pragmatic issues around the length of assessment and practice opportunities would lead to greater acceptance, perhaps as this would facilitate learners’ competence perceptions when faced with a novel situation. Despite this, learners reported that in this situation:


*“the online assessment was a fair assessment using clinical scenarios similar to those we would have been given in an OSCE—given the circumstances I think it was the best option.”*


Although learners indicated suggestions to improve the pragmatics of the OBE assessment experience, they appear to accept the novel method. Learners also accepted the need for assurance that the assessment remains a valid test of ability and made suggestions to do this. Additional support in the learning environment, for example, increasing the availability of practice opportunities including a greater variety in questions may increase the acceptance of OBEs. This may be particularly important for those learners who were less sure that OBEs were a fair test of competence, as highlighted in the earlier quantitative analysis. Taken together these results emphasise that preparing learners for such a test is crucial in enabling them to perceive that they are competent to face the OBE regardless of the prevailing circumstances

## 4. Discussion

This mixed methods investigation examined the relationships between (1) learners self-reported preparations and behaviours in approaching the OBE, (2) their acceptance of the OBE as an authentic and valid test of competence, (3) their endorsement of support from the learning environment, and final examination marks. The quantitative and qualitative results reported indicate high levels of learner acceptance of the implemented OBE, with learners recognising the OBE as a valid test of their competence, and largely supporting the continuation of OBE examinations in future assessment rounds. These understandings were, however, nuanced.

We can firstly consider the relationships between anxiety, preparation, consulting resources and examination performance within this self-reported data set. It would appear that anxiety was a driver for increased time spent in preparation for the examination. However, this increased time did not correlate with better performance. This brings into question the nature and objective of the preparation undertaken, perhaps suggesting a surface learning approach that was ultimately of no benefit in an examination designed to test clinical problem solving. Given that neither preparation time nor anxiety appeared to be correlated with time spent consulting resources in the examination it is interesting to speculate what factors might be driving such behaviour, especially in the face of pre-examination warnings about its likely counter-productive effects. It is possible that a sub-set of learners adopt a very strategic approach to assessment, believing that “gaming” is an effective policy, seemingly encouraged by the unproctored nature of the examination. Furthermore, participants may have self-reported greater levels of preparation than were factual due to the lack of anonymity of their responses. Despite our lack of complete understanding, there do appear to be some important messages to be derived that coincide with an intuitive understanding of student behaviours and which can be used to reinforce guidance to students on their approach to this style of assessment.

Learners endorsed the continuation of the OBE in future assessment rounds. This endorsement increased as learners achieved higher examination marks, and more strongly by learners that endorsed the OBE as a valid test of competence, indicating learner acceptance of the OBE as a suitable alternative assessment method. This echoed the support from learners seen in qualitative analysis. Despite this report, it should be noted that more anxious learners were less likely to endorse the OBE as a valid test of competence and its continuance in future assessment rounds than colleagues reporting lower anxieties. This is perhaps a reflection of mindset and it is of interest whether unmeasured personality factors may have influenced levels of self-efficacy and competence. A number of other studies have reported that learners may prefer open-book to closed-book examinations due to reporting lower anxiety and in making less effort for preparation [11,12,13,14]. Furthermore, increased anxiety may lead to differences in students’ self-assessment of their performance compared to their actual performance [15]. Competence perceptions in learning are thought to be associated with a range of global, contextual and situational factors [16]. However, these same perceptions are well understood to be bolstered by a supportive educational environment that enhances learners’ feelings of autonomy [7,17]. Therefore, considering how to optimise the learning environment surrounding assessment so that learners feel supported, is a design consideration that plays a key part in reassuring learners and their feelings of competence so that they are enabled to successfully manage a situation, in this case, the revised assessment approach. It would have been perfectly possible for these perspectives to have been relegated in favour of expedience during the pandemic, leading to these motivational perspectives being undermined, which could have resulted in a negative impact on performance. Understanding learner perceptions of the environment may be related to acceptance of the assessment method, and support their feelings of competence, which was not assured given the OBEs introduction.

We found, in qualitative analysis, calls for additional practice opportunities, which might, within an appropriate preparatory framework, help learners to understand the nature of the OBE and accept it more readily as a valid test of competence. Learners suggestions for greater practice opportunities might go some way to alleviating the anxieties in a way that supports the acceptance of the OBE and it continuing as part of a suite of assessments. This said, learners were unaware of their assessment marks at the time they participated in the research, however, this may betray learners’ evaluations of the assessment accurately. It should be noted that, despite the opportunity for pre-examination familiarisation and practice, the adopted format was new to the students. It is not possible to determine from this study whether this lack of familiarity was the cause of anxiety rather than some intrinsic aspect of the assessment format. Finally, learners that were more likely to endorse online delivery of teaching also endorsed continued online assessment. The endorsement of continued online delivery was unrelated to other factors measured.

The small sample surveyed here was a limiting factor, nevertheless, the sample was the total population of interest for this investigation. A similar study was carried out anonymously and invited all undergraduate students in all years to participate; the response rate for final year students was 25% [14]. Arguably, final year students might be best informed to self-report their perceptions of OBEs in comparison to closed-book examinations, and the 100% response rate in this study is a positive factor, although it is accepted that there may not be a direct correlation between response rate and validity. Low levels of statistical power may lead to the very real possibility that effect estimates reported are under- or over-inflated within this single centre study, and it may be that extending the sample might have secured a different understanding. However, extending the sample could also have introduced unanticipated confounds. This limitation is acknowledged, as is the exploratory nature of the analysis, and, consequently, the results should be interpreted with a degree of caution. We attempted to support the limits of the qualitative understanding by supplementing it with qualitative data collection and analysis, seeking learners’ views on improvements so that we can design an improved approach that takes account of learner perspectives going forward. The resulting report does this. These challenges aside, we consider that this study contributes to understanding, albeit in a limited manner, to learner perspective on the introduction of OBEs which may, in turn, suggest future research directions. Prior to the examination, strong messages were communicated to the students about the importance of preparation and the inadvisability of over-reliance on freedom to spend a lot of time consulting resources; the design of the examination in promoting the application of knowledge rather than resource-searching skills was reiterated. Most students appear to have taken this onboard but the inter-relationships of perceptions of difficulty, use of resources, anxiety, and performance are worthy of further study.

## 5. Conclusions

Undergraduate dental students are largely agreeable to the OBE approach, arguably providing support for the view that such approaches can be authentic. The results of this study will inform local curricula and assessment development, learning and teaching practices, and have been shared with the student body to support their experience. This study contributes to understanding of undergraduate perceptions on OBEs and encourages dental educators to share best practice.

Research data are not shared due to the requirements of the ethical approval granted.

## Figures and Tables

**Figure 1 dentistry-10-00200-f001:**
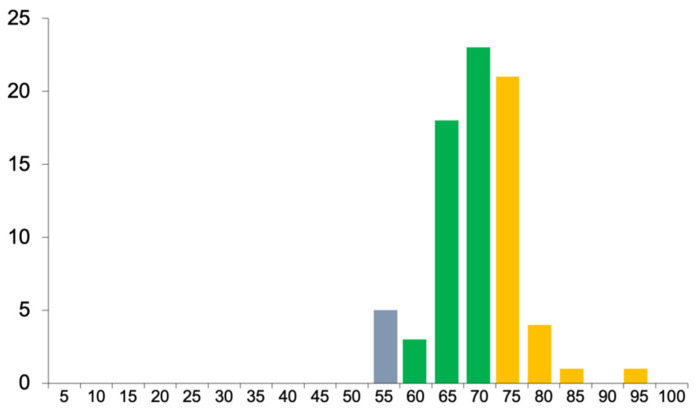
Distribution of final examination marks.

**Figure 2 dentistry-10-00200-f002:**
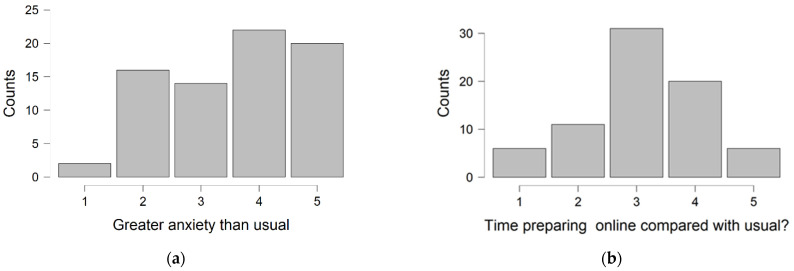
Distribution of participant responses to survey items. (**a**): Greater anxiety than usual; (**b**): Time preparing online than usual; (**c**): Time consulting resources; (**d**): Supportive enviroment; (**e**): Authentic Test; (**f**): Welcome continued online delivery? (**g**): Continue online assessment.

**Table 1 dentistry-10-00200-t001:** Summary of descriptive statistics and Spearman’s correlations (*r_s_*) examining learners’ perceptions of OBE and finals assessment marks.

Item	Median	Range	1	2	3	4	5	6	7	8
1. I felt more anxious about exams than I usually do	4.00	4.00	—							
2. Time preparing online compared with usual	3.00	4.00	0.55 **	—						
3. I spent a lot of time in the exam consulting resources	2.00	4.00	0.10	0.13	—					
4. Supportive learning environment	4.00	2.67	0.05	−0.05	0.04	—				
5. Finals mark	68.00	43.00	−0.14	−0.04	−0.25 *	−0.05	—			
6. The OBE as an authentic test of competence	4.38	3.00	−0.43 **	−0.15	−0.05	0.23	−0.06	—		
7. Welcome continued online delivery?	4.00	4.00	−0.06	−0.05	−0.05	0.16	0.03	0.20	—	
8. Welcome continued online assessment	3.00	4.00	−0.27 *	0.01	−0.02	0.19	−0.27 *	0.42 **	0.39 **	—

* *p* < 0.05, ** *p* < 0.001.

## Data Availability

Data will be provided on request.

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
