# Peer review of "Online Unproctored Clinical Examinations: The Perceptions and Performance of Final Year Dental Students"

_dentistry, 2022, doi:10.3390/dj10110200_

Round 1

Reviewer 1 Report

Dear Authors, this paper about Online unproctored clinical examinations and their perceptions and performance of final year dental students is really interesting and well written.

It can be really helpful for readers who are interested in knowing more about the online learning and all the situation we are currently facing all over the world.

Some issues need to be solved before its final publication in the journal:

Abstract: please divide abstract into chapters: background, methods, results, conclusions

Introduction: introduction in well written but a chapter regarding the impact of the pandemic on dental education. This paper could help: Costa, E. D., Brasil, D. M., Santaella, G. M., Cascante-Sequeira, D., Ludovichetti, F. S., & Freitas, D. Q. (2022). Impact of COVID-19 pandemic on dental education: Perception of professors and students. [Impacto de la pandemia COVID-19 en educación dental: percepción de estudiantes y profesores] Odovtos - International Journal of Dental Sciences, 24(1), 122-133. doi:10.15517/IJDS.2021.46567

Please, in the whole manuscript, references numbers must be written correctly: [2] and not (2).

Materials and methods: do not write just the word “materials”. Write instead “materials and methods”

Ethical approval number 7822, readers do not know what this is, please, explain better were it was obtained and in which date.

Write a single paragraph, the subparagraphs “procedures” and “Data Analysis” are not necessary

Results: results are well written and easy understandable. Figures are out of focus: please fix that

Discussion: in the discussion part some characters are not equals to others, please check it.

Author Response

Thank you for your comments.

Abstract: please divide abstract into chapters: background, methods, results, conclusions - this has been completed as requested. Please confirm whether you wish us to reflect this in the main text (for example, 'background' and 'materials and methods' are requested to be used in the main text).

We have made the remaining edits as suggested to the paper as requested.

We have attached the paper for your review prior to resubmission. Please note this includes minor edits and requests included by the other reviewer.

Reviewer 2 Report

I was pleased to review your work under the title: Online unproctored clinical examinations: the perceptions and performance of final year dental students.

The study is well thought out and written. I suggest correcting the following:

INTRODUCTION:

Much of the introduction is more about material and methods and discussion. For example: "In response to COVID-19", to aims.

MATERIALS AND METHODS: Too long parts. If you already have questions in the table, do not rewrite them in the text.

RESULTS: Also lengthy, but good writing.

DISCUSSION: Other studies are well addressed.

CONCLUSION: Too long. One part of the conclusion is more suitable for discussion. Follow the conclusion through set aims.

Author Response

Thank you for your comments.

We have addressed the suggestions as below, however, we have continued to include in the introduction the section following ''In response to COVID-19'' to aims - as this section provides necessary background regarding the assessment, rather than the materials and methods of the present study. whilst we agree that some of this content could be considered discussion, we feel that its inclusion in the introduction is necessary for the reader to understand the scene in which the study is set, and this has also been considered by the first reviewer as information that clarifies the impact of COVID. We therefore request that the editor consider the final decision in this respect.

We have attached the paper for your review prior to resubmission. Please note this includes minor edits and requests included by the other reviewer.
